# Structure of Machupo virus polymerase in complex with matrix protein Z

Jun Ma[1,4], Shuangyue Zhang[1,2,4] & Xinzheng Zhang [1,2,3 ✉]

The *Arenaviridae* family includes several viruses that cause severe human hemorrhagic fevers with high mortality, with no effective countermeasures currently available. The arenavirus multi-domain L protein is involved in viral transcription and replication and represents a promising target for antiviral drugs. The arenavirus matrix protein Z is a small multi-functional protein that inhibits the activities of the L protein. Here we report the structure of Machupo virus L protein in complex with Z determined by cryo-electron microscopy. The Z protein acts as a staple and binds the L protein with 1:1 stoichiometry at the intersection between the PA-C-like region, RNA-dependent RNA polymerase and PB2-N-like region. Binding of the Z protein may lock the multiple domains of L into a fixed arrangement leading to loss of catalytic activity. These results further our understanding of the inhibitory mechanism of arenavirus replication machinery and provide a novel perspective to develop antiviral drugs.

[1] National Laboratory of Biomacromolecules, CAS Center for Excellence in Biomacromolecules, Institute of Biophysics, Chinese Academy of Sciences, 100101 Beijing, China. [2] University of Chinese Academy of Sciences, 100049 Beijing, China. [3] Center for Biological Imaging, CAS Center for Excellence in Biomacromolecules, Institute of Biophysics, Chinese Academy of Sciences, 100101 Beijing, China. [4] These authors contributed equally: Jun Ma, Shuangyue Zhang. ✉email: xzzhang@ibp.ac.cn

The *Arenaviridae* family is a large group of diverse, enveloped, RNA viruses with negative-sense segmented genomes[1,2]. Several mammalian arenaviruses can infect humans and cause encephalitis and severe hemorrhagic fevers with high mortality and morbidity. Arenaviruses can be divided into two groups[1,2]: Old World arenaviruses including lymphocytic choriomeningitis virus (LCMV) and Lassa virus (LASV); and New World arenaviruses including Junin virus (JUNV) and Machupo virus (MACV).

The arenavirus genome comprises two ambisense RNA segments encoding four viral proteins: immature glycoprotein precursor polyprotein, nucleoprotein (N), large RNA-dependent RNA polymerase (RdRp) (L), and zinc-binding matrix protein (Z)[3–5]. Mature envelope glycoprotein on the viral surface mediates entry into target cells via binding to host receptors. The N protein associates with viral RNA to form the nucleocapsid that protects RNA from cellular nucleases and serves as a template for viral transcription and replication. The L protein is a multidomain, multi-functional protein responsible for transcription and replication of the viral genome[6,7]. The Z protein is a small but multifunctional protein containing fewer than 100 residues[8]. The structure of LASV Z protein has previously been solved by NMR spectroscopy[9] and X-ray crystallography[10] in monomer and dodecamer forms, respectively. These structures revealed that the Z protein N- and C-terminal motifs are flexible, while the central zinc-binding RING motif is responsible for oligomerization. The Z protein mediates multiple functions through interactions with viral and cellular proteins including negative regulation of viral RNA synthesis, orchestration of virus assembly and budding, and countering host immunity[8]. The varying functions of Z protein at different stages of the viral life cycle are likely influenced by transitions between the monomeric and oligomeric forms[10,11].

The arenavirus replication and transcription machinery consists of the L protein and the ribonucleoprotein complex, whose activities are modulated by an inhibitor Z protein[5,12–14]. A previous study has provided structural insight into the L proteins of LASV and MACV and their roles in arenavirus replication[15]. Multiple studies have demonstrated that only the intact RING domain is essential for the inhibitory activity of the Z protein mediated by direct binding to L[11,16–18]. However, the molecular mechanism of inhibition remains unclear because of a lack of structural information on the L–Z complex. In this work, we present the structure of MACV L protein in complex with the Z protein solved by cryo-electron microscopy (cryo-EM). The structure illustrates the negative regulation mechanism of arenavirus L protein through interaction with the Z protein.

## Results

**Structure of the MACV L–Z complex determined by cryo-EM.** Initial attempts to co-express MACV full-length L and Z proteins (Fig. 1a) in insect cells and to obtain a homogeneous Z protein monomer in bacteria failed because of the tendency of the Z protein to form aggregates. Further attempts to express and purify the Z protein in *Escherichia coli* as a fusion to the C-terminus of maltose-binding protein (MBP) yielded sufficient monomeric MBP-Z for structural and biochemical studies (Supplementary Fig. 1a-b). However, most of the fusion protein formed aggregates. The MACV L–Z complex was obtained by incubating the purified L and monomeric MBP-Z proteins at a 1:4 molar ratio for 1 h on ice followed by gel filtration (Fig. 1b, c). The polymerase activity of L could be effectively inhibited by MBP-Z fusion protein (Supplementary Fig. 1c) as reported previously[12].

The structure of the MACV L–Z complex was studied by cryo-EM using a 200 kV Talos Arctica microscope equipped with a K2 camera. The reference-free 2D class averages indicated two forms of particles in the cryo-EM dataset resembling the monomeric and dimeric MACV L protein, respectively. After further data processing, the two structures were reconstructed to overall resolutions at 4.0 and 4.2 Å, respectively, using single-particle methods (Supplementary Figs. 2 and 3, Supplementary Table 1). The final density maps for the monomeric and dimeric L–Z complexes were highly similar to the apo MACV L protein structure[15], except that additional densities were observed that were designated as Z protein in subsequent model building (Fig. 1d). An atomic model of the L protein was generated by fitting the apo MACV L protein structure (PDB 6KLD)[15] into the cryo-EM density map with minor adjustments. An atomic model of Z was built according to the density map based on a homology model of the LASV Z protein[10]. The model of the monomeric L–Z complex was refined using PHENIX and then fitted into the dimeric complex and refined against the map using PHENIX. The C-terminal domain (CTD) of the L protein was only visible in the dimeric complex and was modeled by fitting the CTD of the MACV L–vRNA dimeric complex[15] into the density map because of low resolution.

**Overall structure of the MACV L–Z complexes.** In the monomeric and dimeric MACV L–Z complexes, the stoichiometries of the Z and L proteins were 1:1 and 2:2, respectively (Fig. 2a and Supplementary Fig. 4a). This result was consistent with previous studies suggesting that the Z protein might bind to L in its monomer form or at a molar ratio of 1:1[12]. Structural comparison revealed that the monomeric L–Z complex was nearly identical to one monomer of the dimeric complex (Supplementary Fig. 4b); superposition of the structures yielded root mean square deviations (RMSD) of 0.5 Å, suggesting that contacts between the L and Z proteins were identical in the two structures. The structure of the L protein in the monomeric and dimeric L–Z complexes was also highly similar to those of apo L protein monomer and dimer (Fig. 2b and Supplementary Figs. 4b and 5)[15]; superposition of the structures yielded RMSD of 1.2 and 1.3 Å, respectively, suggesting no major conformational changes following binding of Z. Considering the high similarity of the monomeric and dimeric L–Z complexes, only the monomeric complex is discussed below.

**Structure of MACV Z protein.** In the MACV L–Z complex, only residues 34–82 of the Z protein are visible, which make up the zinc-binding RING domain (Figs. 1a and 2c). The RING domain of MACV Z protein displays all the characteristic features of RING domains[8,10], consisting of an α helix (residues 60–68), two short antiparallel β strands (β1: residues 49–51 and β2: 56–59), two loop regions (loop 1: residues 39–48 and loop 2: residues 69–78), and two conserved zinc-binding sites (Fig. 2c, d). The two zinc fingers, located on opposite sides of RING domain, use conserved C39/C42/C58/C61 and C52/H55/C72/C75 residues to coordinate the two zinc atoms (Fig. 2d), respectively. The N- and C-termini were flexible[9,10] and not resolved in this study. The in vitro pull-down assay of different truncated Z proteins with L showed that RING domain of Z is sufficient for L binding, while deletion of N- and C-termini does not affect the interaction (Supplementary Fig. 6a). This result agrees with previous binding experiments[16,17], which also suggested that the N- and C-termini of Z might not be essential for its inhibitory effects in vitro[16].

Structural comparison between the Z proteins of MACV and LASV showed that the structure of the core RING domain of MACV Z is essentially identical to the reported crystal structure

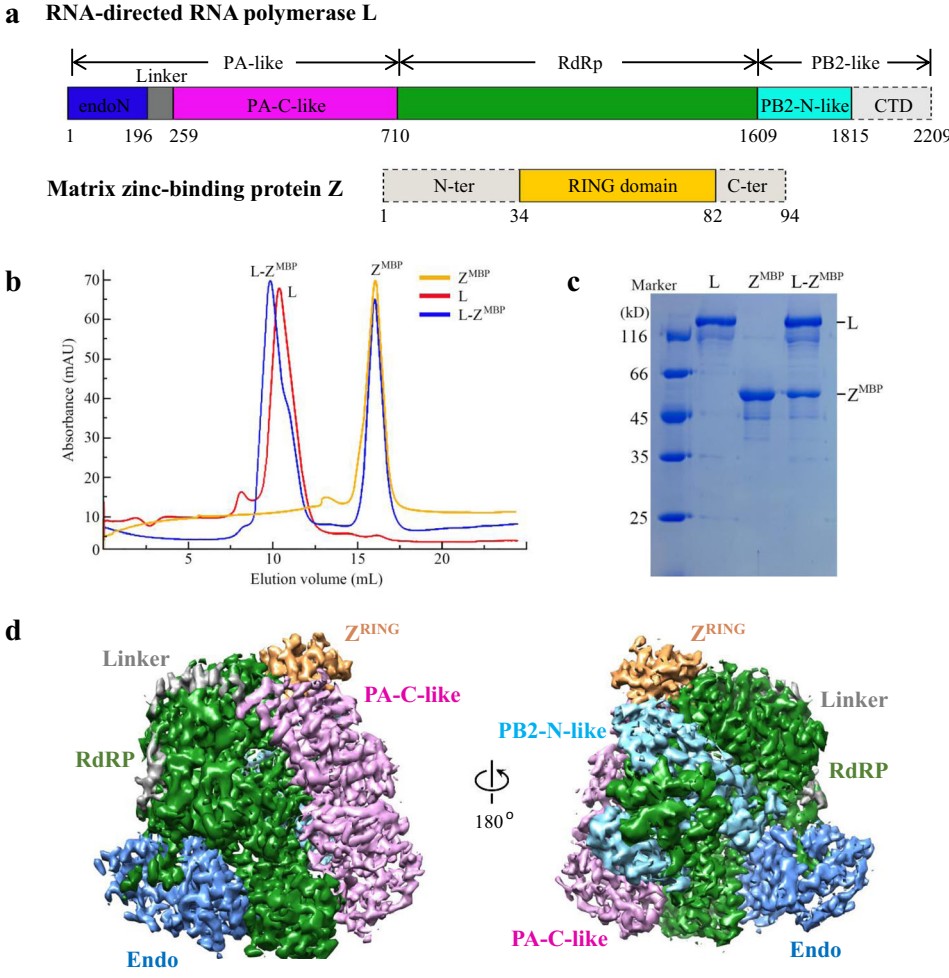

**Fig. 1 Structure determination of the MACV L–Z complex by cryo-EM. a** Schematic diagram of the domain structure of the MACV L and Z proteins. The endoN domain, linker, PA-C-like domain, RdRp, PB2-N-like domain of L are showed in slate, gray, pink, smudge and cyan, respectively, while the solved RING domain of Z colored orange and unresolved regions represented by gray with dotted outlines. The interdomain borders of each protein are labeled with residue numbers. **b** and **c** Gel filtration analysis (**b**) and coomassie-stained SDS–PAGE (**c**) of the L protein, monomeric MBP-Z fusion protein and L–Z$^{MBP}$ complex. Molecular weights (in kilodaltons) of marker are shown on the left, and bands are labeled on the upper and right. The data shown are representative results of more than two independent experiments. **d** Cryo-EM density map of the L–Z monomeric complex determined at 4.0 Å resolution in two orientations. The density map is assembled from the individual domains of L and the RING domain of Z, indicated and colored as in (**a**).

of LASV Z (Supplementary Fig. 6b)[10]. In comparison, the last turn of the helix in LASV Z (residues 58–61) in the NMR structure seems to unfold, suggesting that it could offer more flexibility to the C-terminus.

**The interaction between MACV L and Z proteins**. In the structure of the MACV L–Z complex, monomeric Z protein serves as a staple, interacting with L at the intersection of the core lobe of the PA-C-like region, the palm domain of the RdRp, and the thumb ring of the PB2-N-like region (Fig. 3a, b). Loop 1 of the Z protein contacts the L protein core lobe and the palm domain, while loop 2 interacts with the core lobe and the thumb ring (Fig. 3a, b). The contact regions have a buried area of 958 Å$^2$ in the complex. Residues $_L$Phe642, $_L$Ala643, and $_L$Phe689 of the core lobe and $_L$Phe1378, $_L$Val1388, and $_L$Met1389 of the palm domain form a hydrophobic pocket at the interface between the two domains. The side chains of zTrp43 and zPhe44 are inserted into this hydrophobic pocket, just like one end of the staple (Fig. 3b and Supplementary Fig. 7a). In addition to this hydrophobic interaction, $_Z$Arg36 hydrogen bonds to the main chain of $_L$Asn1179 and the side chain of $_L$Thr1377 from the palm domain of of the RdRp using main chain and side chain, respectively.

$_Z$Cys41 and $_L$Phe689 from the core domain of the PA-C-like region form hydrogen bond through main chains (Fig. 3b). Residues $_Z$His73 and $_Z$Trp76 from Z protein loop 2 form the other end of the staple, with $_Z$His73 makes hydrophobic contact with $_L$Phe688 of the core lobe and $_Z$Trp76 forms hydrophobic contacts with $_L$Asn1712 and $_L$Phe1715 of the thumb ring of the PB2-N-like region (Fig. 3b).

Sequence alignment of multiple *Arenaviridae* family Z proteins indicated that among the residues involved in the interaction between the Z and L proteins, the residues interacting with L in loop 1 are more conserved than those in loop 2 (Supplementary Fig. 7b). In loop 1 of the Z protein, Trp43 is totally conserved in all arenaviruses, while Phe44 is conserved in most arenaviruses and Arg36 is conserved in all the New World arenaviruses. By contrast, only Trp76 in loop 2 is conserved in most of New World Clade B arenaviruses. Intriguingly, a pull-down assay using mutant MACV Z proteins of key residues in L–Z interaction showed that only Trp43Ala completely abolishes the L–Z interaction and PheTrp44Ala remarkably reduces the interaction. Other mutants showed almost no effect on the interaction (Supplementary Fig. 7c). The result is in consistence with the conservation of key residues according to the sequence alignment.

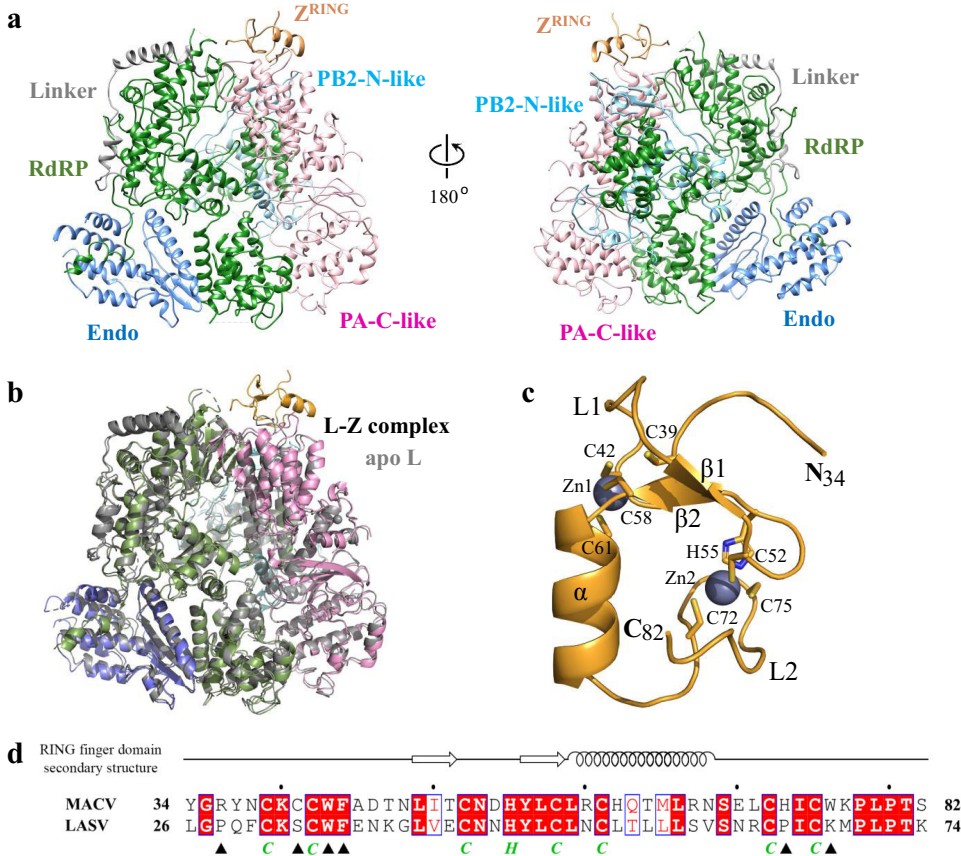

**Fig. 2 Structure of the MACV L–Z complex. a** Atom model of the L–Z complex shown as ribbon in two orientations. The domains are indicated and colored as in Fig. 1a. **b** Superposition of MACV L in the states of apo (colored gray) and Z-bound monomer (colored as Fig. 1a). **c** Structure of Z in the L–Z complex shown as ribbon and colored orange, with two divalent zinc atom shown as gray spheres. Side chains of conserved cysteines and histidine involved in Zn binding are shown as sticks and labeled. Although full-length MBP-Z fusion protein is used for structural study, only residues 34–82 are visible in the complex. **d** Sequence alignment for RING domain between the MACV and LASV Z proteins. Residues involved in the interaction with the L protein are marked below with black solid triangle, while conserved cysteines and histidine involved in Zn binding are marked below with green italicized C or H. Secondary structural elements are depicted on the top of the alignments.

The important role of Trp43 in the L–Z interaction has been reported by previous studies on TCRV and LCMV[11,16,18].

**Conservation analysis of arenavirus L–Z interaction.** A pull-down assay using the Z proteins of various arenaviruses against MACV L protein showed that the Z proteins of New World arenaviruses JUNV, Sabiá virus (SABV), Chapare virus (CHAV), and Tacaribe virus (TCRV) could interact with MACV L with the same capacity as MACV Z, while the Z proteins of Old World arenaviruses (LCMV and LASV) completely lost the binding capacity (Supplementary Fig. 7d). This finding is consistent with the results of a previous study, implying that Z protein cross-inhibits more closely related arenavirus and cannot function as broadly active inhibitors[12]. Although the residues His73 and Trp76 of MACV Z are not conserved in SABV and CHAV, the SABV and CHAV Z proteins behave just like the mutant MACV Z proteins which maintain the L binding capacity.

A model of the LASV L–Z complex obtained by fitting the solved structure of LASV L (PDB 6KLC)[15] and Z (PDB 5I72) proteins[10] into the MACV L–Z complex. In this model, the major end of the staple, comprised of Trp35 and Phe36 of LASV Z protein, is inserted into the hydrophobic pocket mainly formed by residues Tyr649, Val650, Met695 of the core lobe and Phe1381, Val1391, Trp1392 of the palm domain of LASV L protein (Supplementary Fig. 8a, b). This indicates that the residues of LASV L involved in interaction with Z protein,

especially the hydrophobic interaction, had similar biochemical properties to those of MACV L protein (Supplementary Fig. 8c). Thus, the interaction pattern between the L and Z proteins is probably universal among members of the *Arenaviridae* family, although the interaction details may be specific to individual arenavirus clades as a result of independent evolution.

## Discussion

The Z protein has been reported to interact directly with apo L protein and L-vRNA complex[12]. Previous structural study showed that there was no large conformational rearrangement between apo and vRNA-bound MACV L protein, especially in regions required for Z binding[15]. It implies that the Z protein might interact with L in multiple structural or functional states, such as the monomeric and dimeric L protein in both apo and vRNA-bound states. Binding and inhibiting the multiple states of the L protein increase the chance that Z functions the polymerase inhibitory activities, especially when the Z protein under low concentration in vivo. It might be a mechanism for the Z protein to improve the efficiency of inhibition.

Comparing the residues of MACV Z protein involved in the L–Z interaction with residues required for oligomerization[10] and interaction with eIF4E[9] of LASV Z protein, Trp43 and Phe44 of MACV Z protein are involved in all processes (Supplementary Fig. 6c). It indicates that the functions of Z mediated by RING

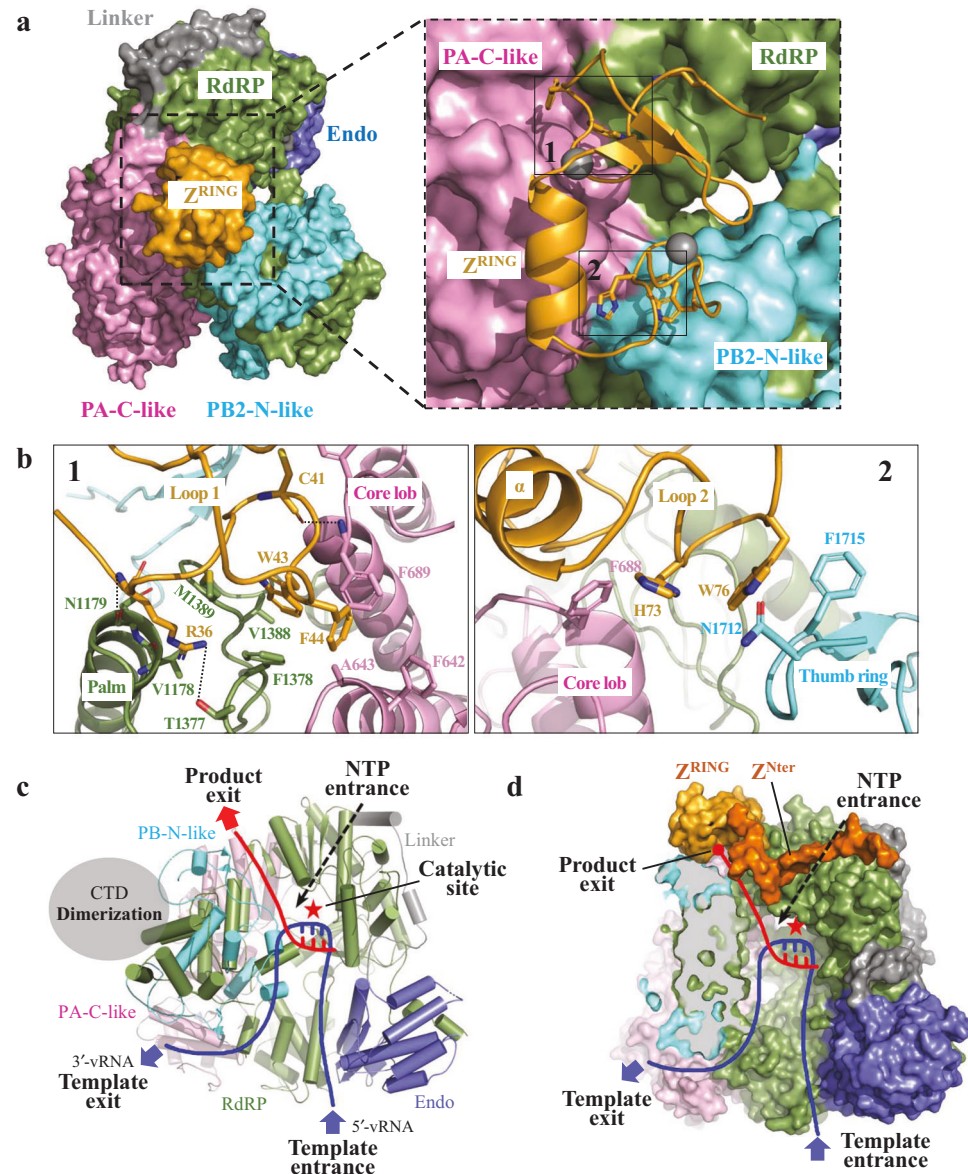

**Fig. 3 Molecular details of the interaction between the Z and L proteins. a** Surface diagram of the L–Z complex with the domains indicated and colored as in Fig. 1a. Inset zoom, the Z protein is represented as ribbon with the two binding sites outlined with black boxes and shown in (**b**). **b** Detailed views of the two interfaces between the Z and L proteins. Left, loop 1 of Z interaction with palm and core lob of the L protein; Right, loop 2 interaction with core lob and thumb ring. The backbones of L and Z are in ribbons representation. Side chains and main chains involved in interactions are shown as sticks and labeled. Oxygen, nitrogen and sulfur atoms shown in red, blue and yellow, respectively, and salt bridges and hydrogen bonds are indicated as black dashed lines. **c** Cartoon model of the L protein showing putative tunnels of the template RNA entrance (blue line) and exit (blue dashed line), the product exit (red line) and NTP entrance (black dashed arrow). The catalytic site is indicated by a red asterisk and the CTD is is represented by a gray oval. **d** Surface representation of L inhibited by Z binding. The RdRP and PB2-N-like domain are partially showed as cut-out view to better display the product RNA exit. The RING domain of Z is colored bright orange, and the modeled N-terminus of MACV Z using LASV Z (PDB 5I72) as template is colored dark orange. The hidden product exit tunnel is showed as red dashed line.

domain are likely mutually exclusive because of steric hindrance in these three states.

Based on the structural information of the RNA polymerases of segmented negative-sense RNA viruses (sNSVs), including MACV L-vRNA structure[15], the pre-initiation, initiation and elongation structures of influenza virus[19,20] and the pre-initiation and elongation structures of La Crosse virus (LACV, *Peribunyaviridae* family)[21,22], we proposed a model of MACV L protein showing the putative tunnels of template RNA entrance/exit, NTP entrance and product RNA exit (Fig. 3c, d). From the proposed model, the Z protein in the MACV L–Z complex is proximal to the putative product RNA exit surrounded by the

structural elements of core lobe, palm domain and thumb ring, about 37.4 Å from the RdRp catalytic triad that is located in the central cavity. Therefore, the location of the Z protein indicates that it unlikely executes polymerase inhibitory activity by directly interfering with the pol catalytic activity, blocking the template RNA entrance/exit tunnels and NTP substrate entry. Considering domain rearrangement is needed for sNSV RNA polymerases to perform and coordinate complex catalytic reactions[6,19–22], the Z protein might exert the inhibitory function by locking the multiple domains of L in a catalytically inactive state. What's more, rather than locking other domains or regions, Z probably specifically prevent the conformational changes of product RNA exit

tunnel by locating at the intersection of the core lobe, palm domain and thumb ring of the L protein as a staple. The polymerase activity of L is inhibited via the blocking of product RNA extension. In addition, the N-terminus of MACV Z protein was modeled using LASV Z (PDB 5I72) as template (Fig. 3d). The flexible N-terminus, especially 10 residues adjacent RING domain[16], is able to approach a helix (residues 1165–1180) of palm domain and two loops (residues 1710–1720 and 1799–1809) of thumb ring, which participate in the formation of the putative product exit tunnel. It might be possible that the N-terminus of Z may also play a role in inhibiting RNA synthesis by blocking the putative product exit tunnel of L.

During the preparation of our revised manuscript, two structural studies of LASV/MACV and JUNV L–Z complex were published by two independent groups[23,24]. JUNV is phylogenetically most closely related to MACV, with the identities of protein sequence of L, Z and RING domain are 74%, 77% and 92%, respectively. Structural comparison of L–Z complex in this study with MACV (PDB 7CKM)[23] and JUNV (PDB 7EJU)[24] showed that these structures are essentially identical, with RMSD of 1.0 and 1.2 Å, respectively (Supplementary Fig. 9). The results and conclusion are basically the same, while differences are not excluded. It is reported[23] that the Z protein of MACV and LASV could mutually interact with and inhibit the activity of the other L protein. In our study, the Z protein of LASV and LCMV did not interact with MACV L. Limited by convincing evidence, no consensus is reached to explain the inhibitory mechanism of Z and therefore further studies are needed.

Effective countermeasures against arenaviruses, such as vaccines and antiviral drugs, are not currently available[25]. The unique strategy used by arenavirus Z protein to inhibit the activity of L provides a novel target for the development of inhibitors. Elucidation of the structural details and molecular mechanisms underlying the L–Z complex formation will be instrumental for structure-based development of inhibitors against arenaviruses, such as engineered Z proteins that more potently inhibit the L protein.

## Methods

**Molecular cloning**. The coding sequences for MACV protein L (Uniprot: Q6IUF8) and Z (Uniprot: Q6IUF9) (Supplementary Table 2) were optimized (GeneralBiol) and synthesized for Spodoptera frugiperda and Escherichia coli, respectively. Protein L was cloned into pFastBac-Dual (Invitrogen) expression plasmid with an C-terminal Twin-Strep-tag under the polyhedron promoter. Sf9 and Hi5 insect cells (Invitrogen) were used for baculovirus propagation and protein expression. Protein Z was cloned into a modified pET22b vector (Novagen) containing an N-terminal 6*His-MBP tag followed by a precission protease (PPase) cleavage site. LASV Z (Uniprot: O73557), LCMV Z (Uniprot: P18541), JUNV Z(Uniprot: Q6IVU5), SABV Z (Uniprot: Q6UY62), CHAV Z (Uniprot: B2C4J2) and TCRV Z (Uniprot: Q88470) were synthesized and and cloned as MACV protein Z.

**Protein expression and purification**. MACV L protein was expressed in Hi5 insect cells (negative for mycoplasma contamination) by the Bac-to-Bac expression system. After 48 h post-infection, cells were collected by centrifugation at $1500 \times g$ for 10 min and resuspended in Lysis Buffer (50 mM Tris–HCl, pH 8.0, 500 mM NaCl, 10% (v/v) glycerol) supplemented with 2 mM DTT and Complete EDTA-free Protease Inhibitor Cocktail (Roche). After lysis by sonication, the lysate was clarified by high-speed centrifugation at $20,000 \times g$ for 60 min at 4 °C. The supernatant was incubated with Strep-Tactin XT (IBA Lifesciences) beads for 1 h at 4 °C. After washing three times with Lysis Buffer to remove non-specifically bound protein, the target protein was subsequently eluted by 2.5 mM D-biotin in Lysis Buffer. The eluted L protein was pooled and further purified using Superdex 200 Increase 10/300 GL column (GE Healthcare) in Gel Filtration Buffer (20 mM Tris–HCl, pH 8.0, 500 mM NaCl and 2 mM DTT). Fractions containing the target protein were concentrated (AmiconUltra, 100 kDa molecular mass cut-off) (Millipore), flash-frozen in liquid nitrogen and stored at −80 °C until further use.

MACV MBP-Z fusion protein was overexpressed in E. coli BL21 (DE3) (Novagen) cells that were induced with 0.3 mM isopropyl-1-thio-β-D-galactopyranoside (IPTG) at $OD_{600} \approx 0.6$ at 16 °C for 16–20 h. Cells were collected by centrifugation at $4000 \times g$ for 20 min and resuspended in Lysis Buffer supplemented with 40 mM imidazole. After lysed by three passes through a

microfluidizer, the lysate was clarified by high-speed centrifugation at $20,000 \times g$ for 60 min at 4 °C. The supernatant was incubated with Ni-NTA agarose (GE Healthcare) for 1 h at 4 °C. Beads were washed three times with with 5 volumes of Lysis Buffer with 40 mM imidazole, and target proteins were eluted in $3 \times 5$ ml Lysis Buffer containing 500 mM imidazole. The eluate was pooled and further purified by size exclusion chromatography using Superdex 200 Increase 10/300 column in Gel Filtration Buffer. Fractions containing monomeric MBP-Z protein were concentrated (AmiconUltra, 30 kDa molecular mass cut-off), flash-frozen in liquid nitrogen and stored at −80 °C until further use. In the preliminary experiment, all the MACV Z protein without MBP tag and the majority of MBP-Z fusion protein formed aggregation, only with a small amount of MBP-Z fusion protein obtained as monomer (Supplementary Fig. 1). However, the monomeric MBP-Z fusion protein was purified to homogeneity for structural study. LASV, LCMV, JUNV, SABV, CHAV and TCRV Z were expressed and purified as full-length wild-type MACV Z.

At each step in the purification procedure, fractions were analyzed by SDS–PAGE. Protein concentration was determined by measuring the absorbance at 280 nm.

**Cryo-EM sample preparation**. For the MACV L–Z complex preparation, purified L protein and MBP-Z fusion protein were mixed with a 1:4 molar ratio and incubated for 1 h on ice. The mixture was further purified by size exclusion chromatography using Superdex 200 Increase 10/300 GL column with Gel Filtration Buffer (Fig. 1b). The fractions of interest were concentrated to high concentration and then diluted to 200 mM NaCl concentration with low-salt buffer before grid preparation.

All Cryo-EM grids were prepared using an automatic plunge freezer Vitrobot Mark IV (FEI) at 16 °C and 80% humidity. An aliquot of 3 μl of the L–Z complex at a concentration of 0.6 mg ml$^{-1}$ was loaded onto a freshly glow-discharged ANTA copper 300 mesh grid[26] with 2 μm holes and 2 μm spacing. After incubation for 10 s, excess sample was blotted with filter paper for 3 s with a blot force of 3, and then the grid was flash-frozen in liquid ethane.

**Cryo-EM image collection and processing**. One cryo-EM dataset of MACV L–Z complex was automatically collected using SerialEM[27] through the beam-image shift data collection method[28] on a 200 kV Tecnai Arctica electron microscope (FEI) equipped with a K2 Summit detector (Gatan). 4907 images were recorded under super-resolution counting mode at a calibrated magnification of ×22,500, resulting in a pixel size of 0.8 Å per pixel and a defocus value ranged from −1.0 to −3.0 μm. Each image was exposed for 6.5 s, with a total exposure dose of ∼60 electrons per Å$^2$ over 32 frames.

Each movie stack was binned and then subjected to beam-induced motion correction and anisotropic magnification correction by the software MotionCor2[29]. This procedure generated two summed images with or without dose weighting for each movie stack. The contrast transfer function (CTF) parameters were estimated by CTFFIND4.1[30] from the summed images without dose-weighting. Further cryo-EM data processing were performed in Relion-3.0.8[31] using the summed images with dose-weighting.

Particles were first automatic picking using Gautomatch (by K. Zhang) and processed with reference-free 2D classification. After the first round of 2D classification, monomeric and dimeric particles were separated, re-extracted with adjusted box-size and subjected to further 2D classification individually. For the monomeric L–Z complex, five 2D class average images of the monomeric L–Z complex were selected as templates for automatic particle picking against the entire dataset. A total of 1,772,578 particles were picked in 4907 micrographs and processed by reference free 2D classification. 890,788 monomeric particles were kept for further 3D classification using cryo-EM map of apo MACV L monomer (EMD-0707)[15] low-pass filtered to 60 Å as the initial model. One among the five classes containing 492,364 particles were selected for further 3D auto-refinement, which resulted in a 4.0 Å density map estimated based on the gold-standard Fourier shell correlation (FSC) with 0.143 criterion (Supplementary Fig. 2). For the dimeric L–Z complex, three 2D class average images of the dimeric L–Z complex were selected as templates for automatic particle picking and 1,015,509 particles were picked. After reference free 2D classification, 280,359 dimeric particles were used for further 3D classification using cryo-EM map of MACV L-vRNA dimer (EMD-0710)[15] as the initial model. The best class among the three containing 149,686 particles were selected for further 3D auto-refinement and resulted in a 4.2 Å density map (Supplementary Fig. 2).

For apo MACV L, one dataset was collected and processed in the same procedure as the MACV L–Z complex. For monomer, 1,489,973 particles were picked in 4078 micrographs and 332,322 monomeric particles were selected for 3D auto-refinement and resulted in a 3.6 Å density map. For dimer, 1,143,890 particles were picked and 52,947 dimeric particles were selected for 3D auto-refinement and resulted in a 5.1 Å density map (Supplementary Fig. 5).

The local resolutions of the final maps were calculated using ResMap[32]. Statistics for image collection and processing are summarized in Supplementary Table 1.

**Model building and refinement**. For the MACV L–Z complex structure, the model of L was generated by manually docking the apo MACV L structure (PDB

6KLD)[15] into the density map using UCSF Chimera[33] and manually fitting the model to the map in Coot[34]. For the Z protein, a homology model of was predicted against LASV Z protein (PDB 5I72) using the Phyre2 server[35] and was successfully placed into the density, which allowed main-chain tracing of RING domain in Coot. The model of the L–Z complex was improved by a few iterative rounds of manual modification in Coot and real-space refinement using PHENIX[36] until no further improvement could be obtained. The models of apo MACV L monomer and dimer were generated by docking the MACV L monomer (PDB 6KLD) and vRNA-bound dimer (PDB 6KLH)[15] into the density maps using UCSF Chimera, respectively. The models were refined iteratively by cycles of real-space refinement using PHENIX and manual modification in Coot until no further improvement could be obtained. The representative densities and atomic models are shown in Supplementary Fig. 3. The refinement statistics for the structural models are summarized in Supplementary Table 1. Structural figures were prepared by either Chimera or PyMOL (https://pymol.org/).

**Pull-down assays**. For pull-down assays, each 30 μg of MBP-Z fusion proteins was mixed with the same amount of MACV L. After one-sixth was taken as input samples, the rest samples were incubated with 25 μl of Amylose Resin for 1 h at 4 °C in Gel Filtration Buffer with a final volume of 500 μl. The beads were washed three times with 500 μl of assay buffer, subsequently incubated with 5 × SDS–PAGE sample buffer at 95 °C for 5 min, separated on 12% SDS–PAGE gels and detected using Coomassie blue staining. Experiments were repeated independently for more than two times.

**In vitro polymerase activity assays**. The assays of polymerase activity inhibited by Z were performed following the procedure below[12,14,15]. Briefly, 10 μM chemically synthesized 19-nt 3′ vRNA and 5′ vRNA (Tsingke) (Supplementary Table 3) were separately denatured at 65 °C for 10 min, and then immediately cooled on ice for use. 1 μM MACV L protein and MBP-Z at specified concentrations were mixed with 1 μM 3′ vRNA and 5′ vRNA in the reaction buffer containing 50 mM Tris–HCl, pH 7.0, 50 mM NaCl, 10 mM KCl, 5 mM MgCl$_2$, 2 mM dithiothreitol and 0.1 mg ml$^{-1}$ bovine serum albumin. After incubation at 25 °C for 30 min, 1 mM ATP/UTP/CTP (Thermo Fisher), 0.12 μCi μl$^{-1}$ [α-$^{32}$P]-GTP (3000 Ci/mmol) (PerkinElmer) and 0.5 U μl$^{-1}$ RiboLock (Thermo Fisher) were added into the mixture to initiate the reaction. Reaction mixtures (20 μl) were incubated at 30 °C for 4 h, terminated by the addition of equal volume of formamide. The products were denatured by heating to 95 °C for 5 min, and 10 μl of each reaction was separated by electrophoresis on denaturing 7 M urea, 20% polyacrylamide gel in 0.5 × TBE buffer. The gels were exposed using a storage phosphor screen and visualized by a Typhoon scanner (GE Healthcare). The RNA marker was labeled with [γ-$^{32}$P]-ATP (PerkinElmer) using T4 polynucleotide kinase (NEB). Experiments were repeated independently for more than two times.

**Reporting summary**. Further information on research design is available in the Nature Research Reporting Summary linked to this article.

## Data availability

The cryo-EM density maps and corresponding coordinates have been deposited in the Electron Microscopy Data Bank (EMDB) and Protein Data Bank (PDB), respectively, under the accession codes: monomeric L–Z complex (EMD-31975, 7VGQ), dimeric L–Z complex (EMD-31983, 7VH1), monomeric apo L (EMD-31985, 7VH3) and dimeric apo L (EMD-31984, 7VH2). All data that support the findings of this study are available from the corresponding authors on reasonable request. Source data are provided with this paper.

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

## Acknowledgements

Cryo-EM data collection was carried out at the Center for Biological Imaging (CBI), Core Facilities for Protein Science at the Institute of Biophysics (IBP), Chinese Academy of Sciences (CAS). We thank Lihong Chen, Boling Zhu, Xujing Li, Xiaojun. Huang and other staff members at the CBI for their support with data collection; and Lingfei Kong for support in sample preparation, cryo-EM data storage and backup. We thank Hongjie Zhang at IBP for the guidance in radioactive polymerase activity assays. The project was funded by the National Key R&D Program of China (2017YFA0504700), the Natural Science Foundation of China (31930069, 31600703), the Strategic Priority Research Program of the Chinese Academy of Sciences (XDB37040101) and the Key Research Program of Frontier Sciences at the Chinese Academy of Sciences (ZDBS-LY- SM003). X.Z. received scholarships from the 'National Thousand Young Talents Program' from the Office of Global Experts Recruitment in China. J.M. is supported by the Youth Innovation Promotion Association of CAS (No. 2019094) and the Beijing Nova Program (No. Z201100006820033).

## Author contributions

J.M. and X.Z. conceived of the project and designed the experiments. J.M. and S.Z. expressed and purified the samples, conducted biochemical studies, prepared the cryo-EM specimens and collected the cryo-EM data. J.M. processed the cryo-EM data and built the model. All authors discussed the experiments, analyzed the data, wrote and approved the manuscript.

## Competing interests

The authors declare no competing interests.
