## [Peer Review File · Nature Communications]

Structure of Machupo virus polymerase in complex with matrix protein ZREVIEWER COMMENTS

Reviewer #1 (Remarks to the Author):

This manuscript describes the structure of the Machupo L-Z polymerase complex. The authors don't provide any rationale as to the mechanism for Z inhibiting L from the structure and based on the similarity to the apo L structure they appear highly similar. An in vitro assay to demonstrate the inhibition of L with the addition of Z would help validate this structure.

Major comments:

Line 137-139: The authors describe that Z protein binds to L far from many sites including the product exit channel but its hard to tell from Fig 3C if that's the case or not.

Lines 173-176: Using "monomeric P" to describe a complex where only a small fragment of the full P protein was used is misleading. It is also what the authors mean by a "pre-initiation configuration" as this could describe either the RdRp active site which is unaffected by P or the configuration of the CD-MT-CTD domains which it is unknown what this configuration represents because no RNA is present. In general, it's not clear what the authors are trying to show by comparing their L-Z structure to the L-P structures of NNS viruses as P and Z are not related and perform completely different functions in the life cycle of these viruses that are only common at the phylum level.

The manuscript could use careful editing to fix many errors.

Minor comments:

Line 45: References are not listed in order of appearance in the manuscript.

Line 61: structures → structure

Fig 1A: not listed in the text

Line 104: "treated" → "discussed"

Line 107: "comprises residues 34-82" → "only residues 34-82 of Z protein are visible"

Line 121: "form bound by RING domains" is this referring to oligomeric Z protein or Z protein bound by other RING domains?

Line 126: Just because the MACV Z protein is unstable during your purification doesn't mean it's also unstable in vivo.

Fig 3C: "Pruduct" → "Product"

Line 207: "Molecule" → "Molecular"

Line 223: "lysed" → "lysis"

Supp Fig 2,4: The local resolution maps appear too noisy, there should be smooth gradients from high to low resolution rather than the splotchy appearance shown here.

Supp Fig 5: It's very hard to tell the difference between the structures being compared.

Reviewer #2 (Remarks to the Author):

Ma et al. reported a cryo-EM structure of the Machupo virus (MACV) L protein in complex with Z protein. MACV is a negative-strand RNA virus. Ma et al. revealed that Z protein acts as a staple and binds the L protein with 1:1 stoichiometry at the intersection between the PA-C-like region, RNA-dependent RNA polymerase, and PB2-N-like region. In this manuscript, Ma et al. also compared the MACV L-Z complex with L-P complexes from several non-segmented negative-sense RNA viruses and highlighted the similarities and differences among them.

Overall, it is a paper with solid experimental data. The reported data unambiguously showed the conclusion. Ma et al. revealed the structural insights between MACV L and Z proteins, highlighting Z's role in the RNA synthesis.

However, there are still some issues regarding the conclusion which need to be addressed:

1. Please remove the citations from the Abstract. Also, spell out the full name for MACV the first time in the Abstract.
2. Line 73. What is the reason to incubate L and MBP-Z proteins at a 1:4 molar ratio, while the L to MBP-Z is at a 1:1 ratio?
3. Line 104. "is treated below"? Did the author mean "discussed"?
4. It is interesting to note that the structural comparison of MACV Z proteins with monomeric LASV Z yielded a high RMSD value than the oligomeric forms of LASV Z. The authors claimed that the monomeric MACV Z is unstable, which may or may not be true. Please provide more evidence.
5. Lines 177-185. The oligomerization domains of P in VSV and RABV are missing, but in RSV and HMPV the oligomerization domains of P are present and adopt a stable structure. Therefore, the argument about the different interaction strategies (inhibitory vs essential cofactor) of Z and P proteins is not convincing. Please clarify.
6. Lines 191-193. It is not clear how Z protein improves the efficiency of inhibition by interacting with L in both apo and vRNA-bound forms. Please explain in more detail.
7. Figure 1, Panel B. The author claimed monomeric MBP-Z fusion protein and L-ZMBP complex in the size column. However, it could not exclude the possibility of the dimeric L-ZMBP complex (2:2 ratio), judging the elution volume.
8. Figure 2, Panel C and D. In the seq alignment, there are only 7 Cys, but 2 Zn²⁺. Is there one missing or using another amino acid (such as His)? Please highlight the Cys residues in panel C.
9. Figure 3, Panel C. It seems Z is far away from the dimerization interface. Does it involve in the dimerization process?
10. Supplementary Figure 1. What is the reason to use PPase to cut the MBP-Z?

Reviewer #3 (Remarks to the Author):

Unlike many enveloped viruses, arenaviruses lack a matrix (M) protein and Z has been shown to play the M role in virion assembly and budding. Z has also been shown to carry out multi-functions during the arenavirus life cycle, including a role in inhibiting viral RNA synthesis by directly interacting with the viral polymerase L. These functions of Z may correlate. Previous NMR studies (PDB ID: 2m1s) showed that Z consists of a central RING finger core domain and flexible N- and C-terminal segments. A crystal structure (PDB ID 5i72) of the RING finger core of Z suggested that this domain could oligomerize (which might be related to its role as a matrix protein). In this manuscript, Ma, et al. described the structure of machupo virus polymerase L in complex with the Z matrix protein, determined by cryoEM to a resolution of 4.0 Å (monomeric form) and 4.5 Å (dimeric form). These structures showed little conformational changes in L upon Z binding, compared to the previously reported structures of apo or vRNA bound L (Peng, et al, Nature 2020) (the authors also reported the apo L structures in this manuscript). Under the authors' experimental conditions, Z bound L in a 1:1 stoichiometry and only the RING finger core was observed in the structures. The authors identified two sets of contacts between L and Z, and proposed a "staple" model of Z binding. Superposing the oligomeric Z to the L-Z structures reported here indicated that other subunit(s) of a Z dimer or oligomer would have collided with the PA-C-like domain of L, suggesting that Z might function as a monomer in the L bound form, although the possibility of L binding to oligomeric Z (such as a Z dimer) could not be ruled out (e.g. if the PA-C like domain of L could rotate out of the way). The location of L where Z binds suggests that it is unlikely that Z directly interferes with the pol catalytic activity, that Z

could directly block substrate entry, and that Z could reach the template entry and exit tunnels. The N- and/or C-termini, which might be the actual functionally relevant parts of Z whereas the RING finger core might simply be an anchor, are not observed in the experimental set up. It is also possible that the RING finger core, through binding to the palm domain of L, might exert the inhibitory function by locking the Pol active site in a catalytically inactive state. These important and most relevant points did not seem to be well discussed or addressed in the manuscript (see section “major points” 2-3).

Overall, this work adds new structural information to the understanding of arenavirus polymerase function (i.e. the inhibitory function of Z in arenaviral genome replication). The quality of the models appears to be reasonable in general, although as pointed out as follows, certain regions require a little bit more careful examination. Although the structures do not clarify many of the unclear questions, the results demonstrate how the two proteins interact. More thorough efforts on relating the structures to the inhibitory function of Z might significantly improve the manuscript and help draw a broad interest in the arenavirus polymerase field and related fields of segmented negative sense RNA virus polymerases. I would like the authors to address a few points as follows. And I would like to particularly point out that more careful and rigorous interpretation and writing would be highly appreciated.

Major points:

1. When describing the L-Z structures and interpreting their functional implications, please be explicit about "far away", e.g. using distance in Ångstroms to describe these (e.g. thanks to the authors for providing the maps and coordinates, I calculated an average distance of 37.42 Å between the SDD catalytic motif and Z). Besides, the rather subjective "far away" statements make little scientific sense without specific functional contexts; in other words, whether it is "far away" depends on given situations and the specific criterion for each case. For example, Z does not seem to be so "far away" from the palm subdomain (see “major point” 2 and 3 for details) and may even engage in direct contacts (e.g. Z residue R36 mainchain amide is 3.5 Å from the carbonyl of the palm residue N1179 and likely forms a hydrogen bond).
2. Functionally, whether direct contacts between the RING finger core of Z and the palm of L, such as what was mentioned above (“major point” 1), could lock the palm to prevent conformational changes that might be required for RNA extension? The authors should check the literatures and/or do structural analysis, and briefly discuss this (please also see “major points” 3).
3. Related to the above points, I superposed the influenza virus initiation and elongation complexes (PDB ID: 6qcs and 6qct) and bunyavirus L bound to vRNAs (PDB ID: 5amq) to the L-Z structures (this study) and noticed that, full-length Z appears to be in proximity to the putative product exit tunnel (although the tunnels of these viruses may not be exactly the same but are good approximation). Particularly, superposition of the NMR structure of Z (PDB ID: 2m1s; full-length Z of Lassa virus) to the L-Z structure showed that the flexible N-terminus of Z (the flexibility might be related to its function) appears to be able to approach a palm helix (residues 1165-1180) and two loops in the PB2-N-like domain (c.a. 1710-1720 and 1799-1809 in the “thumb-ring”). These structural elements seem to line along the “walls” of the putative product exit tunnel and may potentially have implications in the Z function. I tend to disagree with the authors about Z being “far away” from the product exit tunnel. Therefore, I suggest that the authors do more thorough structural analysis using these and other structures, including those in Peng, et al, for more careful interpretation. Z has been shown to inhibit RNA synthesis through directly interacting with L (Kranzusch and Whelan, PNAS 2011), hence relating the structures reported here to the functions, including reasonable and careful speculation, will significantly improve this manuscript. The authors may want to perform assays (or cite published results in the literature, if available) to evaluate the inhibitory function of Z using the full-length vs. the N-terminus deleted versions. And if the authors opt to perform such assays, mutants that would disrupt or partially disrupt the L-Z complex formation (e.g. those mentioned in “major points” 2 and 4) could also be included.
4. I suggest that the authors make more solid, robust and convincing interpretation of the L-Z contacts (I listed some in the “minor points” section). Preferably, the authors could cite or obtain mutagenesis results as evidence.
5. I am not sure about the importance of comparing arenavirus Z proteins to the P proteins of

rhabdoviruses and pneumoviruses. They seem to have rather different functions. The P proteins appear to act as a master regulator (or adaptor) that recruits other viral proteins to carry out various regulatory functions, some of which might be somewhat relevant to those of arenavirus Z proteins. I suggest that rather than dedicate a full section titled "Comparison of MACV L-Z and L-P complexes", the authors focus more on relating the L-Z structures to the function of Z and briefly clarify the L-Z vs. L-P differences in this discussion.

6. The manuscript would benefit significantly from improved writing and additional, very careful editing (and interpretation). On some occasions, I struggled a little attempting to grasp what the authors tried to express. Besides, very importantly, the authors need to use more precise language (as opposed to vague expressions) and be very careful about the terms used, as briefly shown here and in the "minor points" section, which are not an exhaustive list.

Regardless of whether this manuscript is eventually published in this or another journal, I hope the authors could quickly address these and other points so that the results can become available to relevant fields as rigorously and as soon as possible.

Minor points:

1. Line 46: "highly multifunctional" -- simply "multifunctional".
2. Line 49: "intrinsically flexible" -- simply "flexible".
3. Line 79: The resolution for the dimeric form is reported here as 4.5 Å but in the methods and supplementary Figure 2, the best claimed resolution is 4.2 Å. Please be consistent, or if there was a reason, clarify in the methods why the 4.2 Å map was not used as the final map.
4. Line 115: "only the intact RING domain is required for inhibitory activity whereas the N- and C-termini are not essential" -- required for L binding or require for inhibitory activity of Z? Not essential for L binding or not essential for inhibitory activity? These are related but very different concepts. The authors should be very careful when citing previous results and fact check the relevant claims/results.
5. Line 117: The RING finger core in the monomeric and oligomeric forms of Z are very similar. Particularly, considering that these structures (from a different virus, i.e. Lassa virus) were obtained using very different methods at very different resolution, I doubt it is proper to claim "different conformations". In fact, I would argue that they look essentially identical and do not seem to show functionally relevant difference. Besides, RMSD (particularly when the measurements are rather small and close) is nonetheless a rather raw criterion that can be affected by many factors.
6. Line 126: This result did not suggest that monomeric MACV Z protein is "unstable". This only suggested that Z tended to oligomerize whereas the MPB fusion could prohibit oligomerization, likely through structural hindrance of the Z oligomerization interface.
7. The RING finger core of Z seems to form a rather compact structure, as shown by superposition of its L bound form to the published structures obtained by NMR (all states) and X-ray (both subunits in the asymmetrical unit). I suggest the authors be more explicit about "conformational change" (RING finger v.s. termini?).
8. Line 131: simply "The interactions between MACV L and Z proteins".
9. Line 144: these are not "salt bridges"; besides, the contacts are not observed in all models, indicating that the contacts may not be strong. I suggest that the authors examine the models more closely at relevant regions to ensure that interacting residues are built correctly. I also suggest that the authors use restraints so that refinement does not mess up those in poor densities, particularly for refinement in maps at the reported resolution.
10. Line 145: not "enhance this hydrophobic interaction", but rather "in addition to this hydrophobic interaction, we also observed polar contacts between..."
11. Line 148: residue 76 of Z is a tryptophan, not tyrosine. And it does not make "salt bridges".
12. In Figure 3c, please adjust the arrowheads so that their directions are consistent with "in"/"entry" and "out"/"exit". To improve the manuscript (not mandatory), I suggest that the authors use a surface representation (e.g. as an extra side-by-side panel d) to better show the tunnels. Or/and if possible, the authors may want to model an RNA complex into the structure (please note that care must be taken to ensure that proper RNAs are properly fit).
13. Figure S1: please plot the model-to-map cross correlation and the criterion for these plots (i.e. 0.5).

POINT-BY-POINT RESPONSE TO REVIEWER COMMENTS

REVIEWER COMMENTS

Reviewer #1 (Remarks to the Author):

This manuscript describes the structure of the Machupo L-Z polymerase complex. The authors don't provide any rationale as to the mechanism for Z inhibiting L from the structure and based on the similarity to the apo L structure they appear highly similar. An *in vitro* assay to demonstrate the inhibition of L with the addition of Z would help validate this structure.

Response: We thank the reviewer for the comments and helpful suggestion. We have interpreted the inhibitory mechanism of Z protein in detail in the revised manuscript. An *in vitro* polymerase activity assay to verify the inhibitory activity of MBP-Z fusion protein was provided as Supplementary Figure 1c. I would like to apologize that further results are not achieved until submitting the revised manuscript, although we are keeping trying.

Major comments:

Line 137-139: The authors describe that Z protein binds to L far from many sites including the product exit channel but its hard to tell from Fig 3C if that's the case or not.

Response: Thank you for pointing this out. We have replaced the Fig 3c with a modified version and added the panel Fig 3d to further display. In addition, more detailed description and additional discussion about Z protein binding to L was rewritten in Discussion of the revised manuscript.

Lines 173-176: Using "monomeric P" to describe a complex where only a small fragment of the full P protein was used is misleading. It is also what the authors mean by a "pre-initiation configuration" as this could describe either the RdRp active site which is unaffected by P or the configuration of the CD-MT-CTD domains which it is unknown what this configuration represents because no RNA is present.

Response: I would like to apologize for the inaccurate and misleading description. According to the suggestions of the reviewers, we have deleted the section "Comparison of MACV L-Z and L-P complexes".

In general, it's not clear what the authors are trying to show by comparing their L-Z structure to the L-P structures of NNS viruses as P and Z are not related and perform completely different functions in the life cycle of these viruses that are only common at the phylum level.

Response: Thank you very much for pointing this out. The other two reviewers have raised the same question. We have taken the comment into careful consideration and realized that the comparison of molecular mechanism between Z and P protein is not convincing and rigorous. Therefore, we have deleted this section in the revised manuscript.

The manuscript could use careful editing to fix many errors.

Response: Thank you for this suggestion. We have carefully edited the revised manuscript as suggested.

Minor comments:

Line 45: References are not listed in order of appearance in the manuscript.

Response: Thank you for pointing this out. We have fixed this in the revised manuscript.

Line 61: structures → structure

Response: Thank you. We have now corrected this.

Fig 1A: not listed in the text

Response: Thank you. We have now fixed this.

Line 104: “treated” → “discussed”

Response: Thank you. We have now corrected this.

Line 107: “comprises residues 34-82” → “only residues 34-82 of Z protein are visible”

Response: Thank you. We have reworded this sentence as suggested.

Line 121: “form bound by RING domains” is this referring to oligomeric Z protein or Z protein bound by other RING domains?

Response: Thank you for pointing this out. “The form bound by RING domains” is referring to partner-bound Z protein by its own RING domain which including oligomeric Z protein, L-bound Z protein, eIF4E-bound Z protein and so on, as opposed to the free form. According to the suggestions of the other two reviewers, we have rewritten this paragraph and rephrased it .

Line 126: Just because the MACV Z protein is unstable during your purification doesn't mean it's also unstable in vivo.

Response: Thank you for pointing this out. Indeed, it's not suitable to claim that Z protein is unstable. The result only suggested Z tended to oligomerize when MPB tag was removed. Thus, we have rephrased this as follows:

“It suggested that monomeric Z protein may not exist alone and tends to self-assemble or interact with other partner proteins.”

Fig 3C: “Pruduct” → “Product”

Response: Thank you. We have now corrected this.

Line 207: “Molecule” → “Molecular”

Response: Thank you. We have now corrected this.

Line 223: “lysed” → “lysis”

Response: Thank you. We have now corrected this.

Supp Fig 2,4: The local resolution maps appear too noisy, there should be smooth gradients from high to low resolution rather than the splotchy appearance shown here.

Response: Thank you for pointing this out. We have replaced the Supplementary Figure 2e and 4e with a modified version following the advice.

Supp Fig 5: It's very hard to tell the difference between the structures being compared.

Response: Thank you for pointing this out. We have replaced the Supplementary Figure 5 with a modified

version to clearly display the difference.

Reviewer #2 (Remarks to the Author):

Ma et al. reported a cryo-EM structure of the Machupo virus (MACV) L protein in complex with Z protein. MACV is a negative-strand RNA virus. Ma et al. revealed that Z protein acts as a staple and binds the L protein with 1:1 stoichiometry at the intersection between the PA-C-like region, RNA-dependent RNA polymerase, and PB2-N-like region. In this manuscript, Ma et al. also compared the MACV L-Z complex with L-P complexes from several non-segmented negative-sense RNA viruses and highlighted the similarities and differences among them.

Overall, it is a paper with solid experimental data. The reported data unambiguously showed the conclusion. Ma et al. revealed the structural insights between MACV L and Z proteins, highlighting Z's role in the RNA synthesis.

Response: We appreciate the reviewer's very positive comments and valuable suggestions below.

However, there are still some issues regarding the conclusion which need to be addressed:

1. Please remove the citations from the Abstract. Also, spell out the full name for MACV the first time in the Abstract.

Response: Thank you. We have now fixed this as suggested.

2. Line 73. What is the reason to incubate L and MBP-Z proteins at a 1:4 molar ratio, while the L to MBP-Z is at a 1:1 ratio?

Response: Thank you for pointing this out. The L-Z^{MBP} complex was not formed with high efficiency in our study. As the molecular weights of L protein and L-Z^{MBP} complex are very close, it is difficult to completely separate them by size exclusion chromatography using Superdex 200 column. In order to obtain the L-Z^{MBP} complex with minimal unbound L protein, we have screened the molar ratio of L to MBP-Z from 1:1 to 1:8 in the preliminary study. The results showed that the non-bound L protein almost reached the lowest level when the molar ratio was 1:4. A better result couldn't be achieved by increasing the proportion of MBP-Z, while more non-bound L protein appeared under the molar ratio of 1:2. However, excess MBP-Z could be easily isolated from L-Z^{MBP} complex by Superdex 200 column.

3. Line 104. "is treated below"? Did the author mean "discussed"?

Response: Thank you for pointing this out. We have corrected this mistake.

4. It is interesting to note that the structural comparison of MACV Z proteins with monomeric LASV Z yielded a high RMSD value than the oligomeric forms of LASV Z. The authors claimed that the monomeric MACV Z is unstable, which may or may not be true. Please provide more evidence.

Response: Thank you for pointing this out. Structural comparison showed there is slight difference in the helix of RING domain between monomeric and oligomeric forms of Z. MACV Z resembles oligomeric LASV Z with an extended helix and a compact C-terminal tail, yielded a high RMSD value with monomeric LASV Z protein [Supplementary Figure 6b]. I would like to apologize for the inaccurate description "the monomeric MACV Z is unstable". The question has also been raised by the other two reviewers. There is no

more evidence, and the result [Supplementary Figure 1a-b] only suggested Z protein with MPB tag removal tended to oligomerize. Thus, we have rephrased it as follows:

“In our experiments, all the MACV Z protein without MBP tag and the majority of MBP-Z fusion protein forms aggregation, with a small amount of MBP-Z fusion protein obtained as monomer. It suggested that monomeric Z protein may not exist alone and tends to self-assemble or interact with other partner proteins.”

5. Lines 177-185. The oligomerization domains of P in VSV and RABV are missing, but in RSV and HMPV the oligomerization domains of P are present and adopt a stable structure. Therefore, the argument about the different interaction strategies (inhibitory vs essential cofactor) of Z and P proteins is not convincing. Please clarify.

Response: Thank you very much for pointing this out. We have considered this question carefully and realized that it was not convincing and rigorous to compare the molecular mechanism between Z and P protein. Meanwhile, taking into account of the two reviewers' suggestion, we have deleted the section “Comparison of MACV L-Z and L-P complexes”.

6. Lines 191-193. It is not clear how Z protein improves the efficiency of inhibition by interacting with L in both apo and vRNA-bound forms. Please explain in more detail.

Response: Thank you for pointing this out. We have rewritten the sentence in more detail as follows and hope it is clearer:

“It implies that Z protein might interact with L protein in multiple structural or functional states, such as the monomeric and dimeric L protein in both apo and vRNA-bound states. Binding and inhibiting the multiple states of L protein increase the chance that Z functions the polymerase inhibitory activities, especially when Z protein under low concentration in vivo. It might be a kind of mechanism for Z protein to improve the efficiency of inhibition.”

7. Figure 1, Panel B. The author claimed monomeric MBP-Z fusion protein and L-ZMBP complex in the size column. However, it could not exclude the possibility of the dimeric L-ZMBP complex (2:2 ratio), judging the elution volume.

Response: Thank you for pointing this out. We have purified the apo L protein and conducted the structural study [Supplementary Figure 5]. The results of gel-filtration and cryo-EM single particle reconstruction indicated that apo L protein is almost present as monomer, although a small amount of dimeric particles was observed in cryo-EM data, which is in consist with the previous structural study of LASV and MACV L protein [ref#15]. Hence, we didn't consider the L-Z complex as dimer, with the protein peak displayed only a small forward shift against apo L protein. In conclusion, we claimed that L-Z^{MBP} complex was monomer based on the previous study. It would be difficult determine the status of L-Z complex only through the elution volume.

8. Figure 2, Panel C and D. In the seq alignment, there are only 7 Cys, but 2 Zn²⁺. Is there one missing or using another amino acid (such as His)? Please highlight the Cys residues in panel C.

Response: Thank you for pointing this out. The second Zn²⁺ is coordinated by three Cys and one His. We have modified the Fig 2d and Supplementary Figure 7b and highlighted the Cys and His in Fig 2c as suggested. The following sentence has also been added in the revised manuscript:

“The two zinc fingers, located on opposite sides of RING domain, use conserved C39/C42/C58/C61 and C52/H55/C72/C75 to coordinate zinc atoms, respectively.”

9. Figure 3, Panel C. It seems Z is far away from the dimerization interface. Does it involve in the dimerization process?

Response: Thank you for pointing this out. We have replaced the Fig 3c with a modified version. The location of Z protein in L-Z complex shows that Z protein is far away from the dimerization interface, indicates that it is unlikely to involve in the dimerization process. In the gel filtration, no obvious dimeric L-Z complex was observed. The cryo-EM single particle reconstruction [Supplementary Figure 2c & 5c] showed that there is 14% and 23% dimeric particles in apo L protein and L-Z complex, respectively. Compared with 9% and 82% dimeric particles in apo L protein and L-vRNA complex reported [ref#15], there is no solid evidence that Z protein is involved in the dimerization process.

10. Supplementary Figure 1. What is the reason to use PPase to cut the MBP-Z?

Response: Thank you for pointing this out. We initially attempted to obtain homogeneous monomeric Z without tag for structural study. However, the results indicated that almost all the Z protein with tag removal oligomerize or aggregate, without any monomer left. Then we conducted structural study of L-Z complex using MBP-Z fusion protein. In the manuscript, we have proposed that Z may not exist alone and tends to self-assemble or interact with viral or cellular partners. The result that Z formed oligomer after MBP tag removal by PPase will support our proposal.

Reviewer #3 (Remarks to the Author):

Unlike many enveloped viruses, arenaviruses lack a matrix (M) protein and Z has been shown to play the M role in virion assembly and budding. Z has also been shown to carry out multi-functions during the arenavirus life cycle, including a role in inhibiting viral RNA synthesis by directly interacting with the viral polymerase L. These functions of Z may correlate. Previous NMR studies (PDB ID: 2m1s) showed that Z consists of a central RING finger core domain and flexible N- and C-terminal segments. A crystal structure (PDB ID 5i72) of the RING finger core of Z suggested that this domain could oligomerize (which might be related to its role as a matrix protein). In this manuscript, Ma, et al. described the structure of machupo virus polymerase L in complex with the Z matrix protein, determined by cryoEM to a resolution of 4.0 Å (monomeric form) and 4.5 Å (dimeric form). These structures showed little conformational changes in L upon Z binding, compared to the previously reported structures of apo or vRNA bound L (Peng, et al, Nature 2020) (the authors also reported the apo L structures in this manuscript). Under the authors' experimental conditions, Z bound L in a 1:1 stoichiometry and only the RING finger core was observed in the structures. The authors identified two sets of contacts between L and Z, and proposed a “staple” model of Z binding. Superposing the oligomeric Z to the L-Z structures reported here indicated that other subunit(s) of a Z dimer or oligomer would have collided with the PA-C-like domain of L, suggesting that Z might function as a monomer in the L bound form, although the possibility of L binding to oligomeric Z (such as a Z dimer) could not be ruled out (e.g. if the PA-C like domain of L could rotate out of the way). The location of L where Z binds suggests that it is unlikely that Z directly interferes with the pol catalytic activity, that Z could directly block substrate entry, and that Z could reach the template entry and exit tunnels. The N- and/or C-termini, which might be the actual functionally relevant parts of Z whereas the RING finger core might simply be an anchor, are not observed in the experimental set up. It is also possible that the RING finger core, through binding to the

palm domain of L, might exert the inhibitory function by locking the Pol active site in a catalytically inactive state. These important and most relevant points did not seem to be well discussed or addressed in the manuscript (see section “major points” 2-3).

Response: We are very grateful for the constructive comments. We definitely agree with the points the reviewer raised, which are significantly important to our manuscript. We have provided detailed description of L-Z interaction and further discussion about the inhibitory mechanism of Z protein in revised manuscript following the reviewer’s advice.

Overall, this work adds new structural information to the understanding of arenavirus polymerase function (i.e. the inhibitory function of Z in arenaviral genome replication). The quality of the models appears to be reasonable in general, although as pointed out as follows, certain regions require a little bit more careful examination. Although the structures do not clarify many of the unclear questions, the results demonstrate how the two proteins interact. More thorough efforts on relating the structures to the inhibitory function of Z might significantly improve the manuscript and help draw a broad interest in the arenavirus polymerase field and related fields of segmented negative sense RNA virus polymerases. I would like the authors to address a few points as follows. And I would like to particularly point out that more careful and rigorous interpretation and writing would be highly appreciated.

Response: We appreciate the reviewer’s positive comments. We have edited the manuscript extensively following the reviewer’s advice and hope the revised manuscript has addressed the raised points and has been greatly improved.

Major points:

1. When describing the L-Z structures and interpreting their functional implications, please be explicit about "far away", e.g. using distance in Ångstroms to describe these (e.g. thanks to the authors for providing the maps and coordinates, I calculated an average distance of 37.42 Å between the SDD catalytic motif and Z). Besides, the rather subjective "far away" statements make little scientific sense without specific functional contexts; in other words, whether it is “far away” depends on given situations and the specific criterion for each case. For example, Z does not seem to be so "far away" from the palm subdomain (see “major point” 2 and 3 for details) and may even engage in direct contacts (e.g. Z residue R36 mainchain amide is 3.5 Å from the carbonyl of the palm residue N1179 and likely forms a hydrogen bond).

Response: Thanks for this instructive comment. Apologize for the not rigorous and professional description about L-Z structures. We have rephrased the location of Z protein in L-Z complex in the Discussion section.

2. Functionally, whether direct contacts between the RING finger core of Z and the palm of L, such as what was mentioned above (“major point” 1), could lock the palm to prevent conformational changes that might be required for RNA extension? The authors should check the literatures and/or do structural analysis, and briefly discuss this (please also see “major points” 3).

Response: Thanks for this valuable suggestion. We have provided detailed description and further discussion based on the structural analysis and literature. Please refer to the Discussion section in the revised manuscript.

3. Related to the above points, I superposed the influenza virus initiation and elongation complexes (PDB ID: 6qcs and 6qct) and bunyavirus L bound to vRNAs (PDB ID: 5amq) to the L-Z structures (this study) and noticed that, full-length Z appears to be in proximity to the putative product exit tunnel (although the tunnels of these viruses may not be exactly the same but are good approximation). Particularly, superposition of the

NMR structure of Z (PDB ID: 2m1s; full-length Z of Lassa virus) to the L-Z structure showed that the flexible N-terminus of Z (the flexibility might be related to its function) appears to be able to approach a palm helix (residues 1165-1180) and two loops in the PB2-N-like domain (c.a. 1710-1720 and 1799-1809 in the “thumb-ring”). These structural elements seem to line along the “walls” of the putative product exit tunnel and may potentially have implications in the Z function. I tend to disagree with the authors about Z being “far away” from the product exit tunnel. Therefore, I suggest that the authors do more thorough structural analysis using these and other structures, including those in Peng, et al, for more careful interpretation. Z has been shown to inhibit RNA synthesis through directly interacting with L (Kranzusch and Whelan, PNAS 2011), hence relating the structures reported here to the functions, including reasonable and careful speculation, will significantly improve this manuscript. The authors may want to perform assays (or cite published results in the literature, if available) to evaluate the inhibitory function of Z using the full-length vs. the N-terminus deleted versions. And if the authors opt to perform such assays, mutants that would disrupt or partially disrupt the L-Z complex formation (e.g. those mentioned in “major points” 2 and 4) could also be included.

Response: We are grateful for the reviewer’s careful and thorough review of the manuscript, as well as lots of constructive suggestions. Following the reviewer’s advice, we have proposed a model of MACV L protein showing the putative tunnels of template RNA entrance/exit, NTP entrance and product RNA exit, using the structural information of MACV L-vRNA complex, the pre-initiation, initiation and elongation complexes of influenza virus and the pre-initiation and elongation structures of LACV. Based on the predicted model, we proposed that Z protein probably specifically prevent the conformational changes of product RNA exit tunnel which is required for RNA extension. In addition, the predicted model of N terminus of MACV Z protein indicated that it is possible that Z protein blocks the putative product exit tunnel using the RING domain and N-terminus together. For details, please refer to the new added paragraphs in the Discussion section.

During the writing, the previous studies and results were cited rigorously, and thorough structural analysis, careful interpretation and reasonable speculation were carried out. However, I would like to apologize that *in vitro* polymerase activity assays to evaluate the inhibitory function of Z protein was not successful until submitting the revised manuscript, although we are keeping trying. There are still many questions unclear to the inhibitory mechanism of Z protein, which we hope to clarify in the future. After revising the manuscript following the reviewer’s comments, our manuscript has been greatly improved. We would like to thank you once again.

4. I suggest that the authors make more solid, robust and convincing interpretation of the L-Z contacts (I listed some in the “minor points” section). Preferably, the authors could cite or obtain mutagenesis results as evidence.

Response: Thank you for this valuable suggestion. We have rewritten the two sections “The interaction between MACV L and Z proteins” and “Conservation analysis of arenavirus L-Z interaction”, in hoping to have provided convincing interpretation of the L-Z contacts. In addition, the results of mutagenesis and reference citations are included in the revised manuscript as suggested.

5. I am not sure about the importance of comparing arenavirus Z proteins to the P proteins of rhabdoviruses and pneumoviruses. They seem to have rather different functions. The P proteins appear to act as a master regulator (or adaptor) that recruits other viral proteins to carry out various regulatory functions, some of which might be somewhat relevant to those of arenavirus Z proteins. I suggest that rather than dedicate a full

section titled “Comparison of MACV L-Z and L-P complexes”, the authors focus more on relating the L-Z structures to the function of Z and briefly clarify the L-Z vs. L-P differences in this discussion.

Response: Thank you for this suggestion. Reviewer #1 has given the same comment. We have taken the comment into careful consideration and realized that the comparison of molecular mechanism between Z and P protein is not convincing and rigorous. Hence, we have deleted the section “Comparison of MACV L-Z and L-P complexes” and focus more on the relationship between the structure and mechanism of Z protein.

6. The manuscript would benefit significantly from improved writing and additional, very careful editing (and interpretation). On some occasions, I struggled a little attempting to grasp what the authors tried to express. Besides, very importantly, the authors need to use more precise language (as opposed to vague expressions) and be very careful about the terms used, as briefly shown here and in the “minor points” section, which are not an exhaustive list.

Regardless of whether this manuscript is eventually published in this or another journal, I hope the authors could quickly address these and other points so that the results can become available to relevant fields as rigorously and as soon as possible.

Response: Thank you for this valuable suggestion. I would like to apologize for not careful and rigorous interpretation and writing. We have edited the manuscript extensively as suggested and hope that the revised manuscript is easier to read and understand.

Minor points:

1. Line 46: "highly multifunctional" -- simply "multifunctional".

Response: Thank you. We have now corrected this.

2. Line 49: “intrinsically flexible” -- simply “flexible”.

Response: Thank you. We have now corrected this.

3. Line 79: The resolution for the dimeric form is reported here as 4.5 Å but in the methods and supplementary Figure 2, the best claimed resolution is 4.2 Å. Please be consistent, or if there was a reason, clarify in the methods why the 4.2 Å map was not used as the final map.

Response: Apologize for the mistake. The resolution for the dimeric L-Z complex is 4.2 Å. We have corrected this typo in the revised manuscript.

4. Line 115: “only the intact RING domain is required for inhibitory activity whereas the N- and C-termini are not essential” -- required for L binding or require for inhibitory activity of Z? Not essential for L binding or not essential for inhibitory activity? These are related but very different concepts. The authors should be very careful when citing previous results and fact check the relevant claims/results.

Response: Thank you for this suggestion. We apologize for not clearly distinguishing the concepts between “binding” and “inhibitory” in the manuscript. We have carefully checked and cited the previous studies, and rewritten this sentence as follows and hope it is clearer:

“The results are in agreement with multiple studies that the intact RING domain of Z protein is required for L binding and its inhibitory activity whereas the N- and C-termini may be not essential for binding and inhibitory.”

5. Line 117: The RING finger core in the monomeric and oligomeric forms of Z are very similar. Particularly, considering that these structures (from a different virus, i.e. Lassa virus) were obtained using very different methods at very different resolution, I doubt it is proper to claim “different conformations”. In fact, I would argue that they look essentially identical and do not seem to show functionally relevant difference. Besides, RMSD (particularly when the measurements are rather small and close) is nonetheless a rather raw criterion that can be affected by many factors.

Response: Thank you for pointing this out. Structural comparison showed there is slight difference in the helix of RING domain [Supplementary Figure 6b] between monomeric and oligomeric forms of Z. However, no functional evidence reveals the biological significance of the difference. It will be not convincing and rigorous to claim “different conformations” or “conformational change”. Thus, we have rephrased “conformational change” and rewritten the relevant paragraphs to make sure the description is accurate in the revised manuscript:

“... .. implying structural plasticity of RING domain of Z protein under different states.”

6. Line 126: This result did not suggest that monomeric MACV Z protein is “unstable”. This only suggested that Z tended to oligomerize whereas the MPB fusion could prohibit oligomerization, likely through structural hindrance of the Z oligomerization interface.

Response: Thank you for pointing this out. Indeed, we did not precisely express the description about Z protein. We have rephrased it as follows:

“In our experiments, all the MACV Z protein without MBP tag and the majority of MBP-Z fusion protein forms aggregation, with a small amount of MBP-Z fusion protein obtained as monomer. It suggested that monomeric Z protein may not exist alone and tends to self-assemble or interact with other partner proteins.”

7. The RING finger core of Z seems to form a rather compact structure, as shown by superposition of its L bound form to the published structures obtained by NMR (all states) and X-ray (both subunits in the asymmetrical unit). I suggest the authors be more explicit about “conformational change” (RING finger v.s. termini?).

Response: Thank you for this suggestion. We have rephrased “conformational change” and rewritten the relevant paragraphs more precisely in the revised manuscript.

8. Line 131: simply "The interactions between MACV L and Z proteins".

Response: Thank you. We have rewritten this sentence following the advice.

9. Line 144: these are not “salt bridges”; besides, the contacts are not observed in all models, indicating that the contacts may not be strong. I suggest that the authors examine the models more closely at relevant regions to ensure that interacting residues are built correctly. I also suggest that the authors use restraints so that refinement does not mess up those in poor densities, particularly for refinement in maps at the reported resolution.

Response: Thank you for this valuable suggestion. We have examined and refined the structure carefully as suggested to ensure that the model are built correctly according to the density map. The mistake has been corrected and the description of the interaction was rewritten as follows:

“In addition to this hydrophobic interaction, z Arg36 hydrogen bonds to the main chain of l Asn1179 and the side chain of l Thr1377 from the palm domain of of the RdRp using main chain and side chain, respectively. z Cys41 and l Phe689 from the core domain of the PA-C-like region form hydrogen bond through main chains. ”

10. Line 145: not “enhance this hydrophobic interaction”, but rather "in addition to this hydrophobic interaction, we also observed polar contacts between..."

Response: Thank you for this suggestion. We have reworded this sentence following the advice.

11. Line 148: residue 76 of Z is a tryptophan, not tyrosine. And it does not make “salt bridges”.

Response: Thank you. We have corrected this mistake.

12. In Figure 3c, please adjust the arrowheads so that their directions are consistent with “in”/“entry” and “out”/“exit”. To improve the manuscript (not mandatory), I suggest that the authors use a surface representation (e.g. as an extra side-by-side panel d) to better show the tunnels. Or/and if possible, the authors may want to model an RNA complex into the structure (please note that care must be taken to ensure that proper RNAs are properly fit).

Response: Thank you for this valuable suggestion. We have replaced the Fig 3c with a modified version and added the panel Fig 3d using surface representation as suggested. The putative tunnels of template, product and substrate are modeled, however not using the real RNA. In addition, more detailed description and additional discussion about this model was rewritten in the Discussion section of the revised manuscript.

13. Figure S1: please plot the model-to-map cross correlation and the criterion for these plots (i.e. 0.5).

Response: Thank you for your suggestion. Are you referring to the Supplementary Figure 2 & 4? The FSC plot of the model-to-map has been provided with resolutions indicated at FSC = 0.5.

REVIEWER COMMENTS

Reviewer #2 (Remarks to the Author):

This revised submission has successfully addressed most of the concerns I raised in the previous review. I do not have more questions now.

Reviewer #3 (Remarks to the Author):

I appreciate the authors for addressing the questions in the revised manuscript. The authors provided polymerase activity assays demonstrating that increased amounts of Z seemed to decrease RNA synthesis. The authors included pull down assays showing the effects of mutations in key residues involved in the L-Z interaction. The authors removed the entire section of the inappropriate comparison to the L-P complex of pneumovirus and rhabdoviruses. And the authors provided model-to-map FSC curves. The authors reorganized or rephrased some of the text but need more careful wording to make rigorous claims. Particularly I suggest that the authors pay attention to what the data really suggested, making sure that they are not over-claiming. The authors can improve the manuscript by moving technical details in the main text into the methods/supplemental information. Moreover, I suggest the authors depict the figures more explicitly while keeping the descriptions short and concise -- please note that this applies to the captions of many figures. I hope that the authors could dedicate time and efforts on these and other issues and get a final, well-written version suitable for publication as soon as possible. Specific points (not an exhaustive list) are as follows.

Line 94 and 98: "superimposition" -> "superposition"

Line 109: "The N- and C-termini were not resolved in the structure, probably because of their intrinsic high flexibility as previously suggested^{9,10}." -> "The N- and C-termini were flexible^{9,10} and were not resolved in this study."

Line 113-115: The data from the authors only suggested that the truncation of termini did not affect binding of Z to L but did not suggest whether or not the inhibitory function was affected. I suggest that the author clarify this with statements such as "This result agrees with previous binding experiments^{16,17}, which also suggested that the N- and C-termini of Z might not be essential for its inhibitory effects in vitro¹⁶." If the authors insist that they want to claim that the inhibitory function is also affected, the authors must perform polymerase activity assays using FL vs. truncated versions of the Z protein.

Line 116-124: The discussion on "plasticity" (which the authors changed from "difference in conformations") does not seem to offer much significance to the manuscript. I suggest that the authors simply state the fact that "The structure of the core RING finger domain of Z is essentially identical to the reported crystal structure. In comparison, the last turn of the helix in Lassa virus Z (residues 58-61) in the NMR structure seems to unfold, suggesting that it could offer more flexibility to the C-terminus." Besides, there does not seem to be enough evidence to attribute such a difference solely to monomeric vs. oligomeric "states" of Z. It is probably worth to point out the fact that the different structure in the C-terminal region of helix 1 was determined using NMR.

Line 118: "a compact C-terminal tail" is not clear. Did the authors refer to the C-terminus of Z or the C-terminal end of the core domain? The entire core domain of the RING finger looks compact while peripheral regions next to the N- and C-termini are more mobile as expected. Please be explicit to avoid confusion.

Line 121: (related to above) I suggest that the aggregation problem the authors observed goes into methods as aggregation of the recombinantly expressed Z could be due to many reasons. It is well known that Z tends to oligomerize or to bind to partner proteins and hence this does not need such a "suggestion", which, rather than adds any significance, seems to weaken the argument.

Line 135: "Hydrophobic residues of the Z protein (zTrp43 and zPhe44) are inserted into the hydrophobic pocket like one end of a staple." -> "Side chains of zTrp43 and zPhe44 are inserted into this hydrophobic pocket."

Line 174: "Arenaviridae" -> "Arenaviridae"

Line 184: delete "kind of"; "Z protein" -> "Z" or "the Z protein" (wherever applicable)

Line 184: "Z protein" -> "Z" or "the Z protein"; "MACV L-Z complex" -> "the MACV L-Z complex"; "proximity" -> "proximal" or "in proximity"

Line 198: "...thumbing." -> "...thumbing, about 37.4 Å from the RdRp catalytic triad that is located in the central cavity."

Line 207: "N-terminus of MACV Z protein was predicted against LASV" – I am not sure what the authors mean by "against". Does the author mean that "superposition of the MACV Z structure to our model indicates that the flexible N terminus of Z might interfere with **** of L"?

Line 211: "It cannot be excluded that Z protein blocks the putative product exit tunnel using the RING domain and N-terminus together." -> "It might be possible that the N-terminus of Z may also play a role in inhibiting RNA synthesis by blocking the putative product exit tunnel of L."

Line 220: "Z protein" -> "the Z proteins"; "didn't" -> "did not".

Line 220-223: "For JUNV, dimeric L protein was not observed²⁴ which is present in MACV^{15,23}. In addition, the interpretation about the inhibitory mechanism of Z protein was slightly different, probably due to the lack of solid data. Further research is needed to accurately clarify the regulatory mechanism of Z protein." -> "Limited by convincing evidence, no consensus is reached to explain the inhibitory mechanism of Z and therefore further studies are needed."

Fig. 2c. The rotamers of some of the zinc coordinating residues seem to be incorrect. Please fix the geometry in the model and remake the figure using the re-refine coordinates.

Fig. 3c. Please clip the surface representation open to reveal the tunnels (something like Fig 2b-c or Fig 3b in ref. 18 of the initial manuscript). The authors may need to adjust the orientation and to make sure Fig. 3b assumes the same orientation. Besides, the authors may not want to clip Z (surface or cartoon representation).

Suppl. Fig. 1c. needs better (and concise) description. Please also depict more explicitly for some other figures.

Suppl. Fig. 6b. I suggest that the authors clarify the methods used for structure determination, e.g. "LASV, 2MIS (NMR)" so that readers can make their own judgment as to whether some of the differences were due to methods or real differences.

Suppl. Fig. 7. Figure legends do not match panels (B and C swapped, and that for panel D is missing?).

REVIEWERS' COMMENTS

Reviewer #2 (Remarks to the Author):

This revised submission has successfully addressed most of the concerns I raised in the previous review. I do not have more questions now.

Response: We thank the reviewer for the positive response.

Reviewer #3 (Remarks to the Author):

I appreciate the authors for addressing the questions in the revised manuscript. The authors provided polymerase activity assays demonstrating that increased amounts of Z seemed to decrease RNA synthesis. The authors included pull down assays showing the effects of mutations in key residues involved in the L-Z interaction. The authors removed the entire section of the inappropriate comparison to the L-P complex of pneumovirus and rhabdoviruses. And the authors provided model-to-map FSC curves. The authors reorganized or rephrased some of the text but need more careful wording to make rigorous claims. Particularly I suggest that the authors pay attention to what the data really suggested, making sure that they are not over-claiming. The authors can improve the manuscript by moving technical details in the main text into the methods/supplemental information. Moreover, I suggest the authors depict the figures more explicitly while keeping the descriptions short and concise -- please note that this applies to the captions of many figures. I hope that the authors could dedicate time and efforts on these and other issues and get a final, well-written version suitable for publication as soon as possible. Specific points (not an exhaustive list) are as follows.

Response: We thank the reviewer for the positive response to the revised manuscript. We have carefully considered the remaining comments and revised the manuscript as suggested. Some technical content has been moved to the methods, the captions of some figures has been reworded. We hope that the revision should have addressed the reviewer's concerns, which helped to greatly improve the manuscript. We would like to thank reviewer once again.

Line 94 and 98: "superimposition" -> "superposition"

Response: Thank you. We have now corrected this.

Line 109: "The N- and C-termini were not resolved in the structure, probably because of their intrinsic high flexibility as previously suggested^{9,10}." -> "The N- and C-termini were flexible^{9,10} and were not resolved in this study."

Response: Thank you. We have reworded this sentence as suggested.

Line 113-115: The data from the authors only suggested that the truncation of termini did not affect binding of Z to L but did not suggest whether or not the inhibitory function was affected. I suggest that the author clarify this with statements such as "This result agrees with previous binding experiments^{16,17}, which also suggested that the N- and C-termini of Z might not be essential for its inhibitory effects in vitro¹⁶." If the authors insist that they want to claim that the inhibitory

function is also affected, the authors must perform polymerase activity assays using FL vs. truncated versions of the Z protein.

Response: Thank you for the critical comment. We agree that it is very speculative to claim that the truncation of termini of Z affects the inhibitory function without additional evidence. We have reworded this sentence following your advice.

Line 116-124: The discussion on “plasticity” (which the authors changed from “difference in conformations”) does not seem to offer much significance to the manuscript. I suggest that the authors simply state the fact that “The structure of the core RING finger domain of Z is essentially identical to the reported crystal structure. In comparison, the last turn of the helix in Lassa virus Z (residues 58-61) in the NMR structure seems to unfold, suggesting that it could offer more flexibility to the C-terminus.” Besides, there does not seem to be enough evidence to attribute such a difference solely to monomeric vs. oligomeric “states” of Z. It is probably worth to point out the fact that the different structure in the C-terminal region of helix 1 was determined using NMR.

Response: Thank you for the critical comments. With careful consideration, we completely agree that there is no enough evidence to attribute the structural difference of Z to its difference “states”. It is also very speculative that the “plasticity” of Z plays a role in its functions, offering no much significance to our manuscript. Therefore, we have reworded this sentence following your advice.

Line 118: “a compact C-terminal tail” is not clear. Did the authors refer to the C-terminus of Z or the C-terminal end of the core domain? The entire core domain of the RING finger looks compact while peripheral regions next to the N- and C- termini are more mobile as expected. Please be explicit to avoid confusion.

Response: We are sorry for this vague expression. “a compact C-terminal tail” is referred to the C-terminal end of the core RING finger domain. Following your advice, we have reworded this sentence as in the previous comment.

Line 121: (related to above) I suggest that the aggregation problem the authors observed goes into methods as aggregation of the recombinantly expressed Z could be due to many reasons. It is well known that Z tends to oligomerize or to bind to partner proteins and hence this does not need such a “suggestion”, which, rather than adds any significance, seems to weaken the argument.

Response: Thank you for this valuable suggestion. We agree with the reviewer that it might be not reasonable to suggest the related functions based on the aggregation problem observed during protein expression and purification. We have moved this part into the Methods section and reworded the sentences as suggested.

Line 135: “Hydrophobic residues of the Z protein (zTrp43 and zPhe44) are inserted into the hydrophobic pocket like one end of a staple.” -> “Side chains of zTrp43 and zPhe44 are inserted into this hydrophobic pocket.”

Response: Thank you. We have reworded this sentence as follows:

“The side chains of zTrp43 and zPhe44 are inserted into this hydrophobic pocket, just like one end of the staple.”

Line 174: “Arenaviridae” -> “Arenaviridae”

Response: Thank you. We have now corrected this.

Line 184: delete “kind of”; “Z protein” -> “Z” or “the Z protein” (wherever applicable)

Response: Thank you. We have now corrected these.

Line 184: “Z protein” -> “Z” or “the Z protein”; “MACV L-Z complex” -> “the MACV L-Z complex”; “proximity” -> “proximal” or “in proximity”

Response: Thank you. We have now fixed these.

Line 198: “...thumb ring.” -> “...thum bring, about 37.4 Å from the RdRp catalytic triad that is located in the central cavity.”

Response: Thank you. We have added this sentence in the revised manuscript.

Line 207: “N-terminus of MACV Z protein was predicted against LASV” – I am not sure what the authors mean by “against”. Does the author mean that “superposition of the MACV Z structure to our model indicates that the flexible N terminus of Z might interfere with **** of L”?

Response: We are sorry for the vague expression. We have reworded this sentence as follows:

“In addition, the N-terminus of MACV Z protein was modeled using LASV Z (PDB 5I72) as template.”

Line 211: “It cannot be excluded that Z protein blocks the putative product exit tunnel using the RING domain and N-terminus together.” -> “It might be possible that the N-terminus of Z may also play a role in inhibiting RNA synthesis by blocking the putative product exit tunnel of L.”

Response: Thank you. We have reworded this sentence as suggested.

Line 220: “Z protein” -> “the Z proteins”; “didn’t” -> “did not”.

Response: Thank you. We have now corrected this .

Line 220-223: “For JUNV, dimeric L protein was not observed²⁴ which is present in MACV^{15,23}. In addition, the interpretation about the inhibitory mechanism of Z protein was slightly different, probably due to the lack of solid data. Further research is needed to accurately clarify the regulatory mechanism of Z protein.” -> “Limited by convincing evidence, no consensus is reached to explain the inhibitory mechanism of Z and therefore further studies are needed.”

Response: Thank you. We have reworded this sentence as suggested.

Fig. 2c. The rotamers of some of the zinc coordinating residues seem to be incorrect. Please fix the geometry in the model and remake the figure using the re-refine coordinates.

Response: Thank you for pointing this out. We have refined the model and replaced the panel with a modified version.

Fig. 3c. Please clip the surface representation open to reveal the tunnels (something like Fig 2b-c or Fig 3b in ref. 18 of the initial manuscript). The authors may need to adjust the orientation and to make sure Fig. 3b assumes the same orientation. Besides, the authors may not want to clip Z (surface or cartoon representation).

Response: Thank you for this suggestion. We have modified the panel as suggested to better display the product RNA exit which is blocked by the RING domain and the N terminal of the Z protein.

Suppl. Fig. 1c. needs better (and concise) description. Please also depict more explicitly for some other figures.

Response: Thank you for this suggestion. We have reworded the description about Suppl. Fig. 1c, as well as other figures.

Suppl. Fig. 6b. I suggest that the authors clarify the methods used for structure determination, e.g. “LASV, 2MIS (NMR)” so that readers can make their own judgment as to whether some of the differences were due to methods or real differences.

Response: Thank you. We have modified the panel as suggested.

Suppl. Fig. 7. Figure legends do not match panels (B and C swapped, and that for panel D is missing?).

Response: Thank you for pointing this out. We have fixed the mistake in the revised manuscript.